# Wildfires and Older Adults: A Scoping Review of Impacts, Risks, and Interventions

**DOI:** 10.3390/ijerph20136252

**Published:** 2023-06-29

**Authors:** Colleen Cummings Melton, Carson M. De Fries, Rebecca M. Smith, Lisa Reyes Mason

**Affiliations:** Graduate School of Social Work, University of Denver, Denver, CO 80210, USA; colleen.cummings@du.edu (C.C.M.); carson.defries@du.edu (C.M.D.F.); rebecca.m.smith@du.edu (R.M.S.)

**Keywords:** wildfires, climate change, disaster recovery, evacuation, adaptation, mitigation, older adults, elders

## Abstract

Climate change is leading to worsening disasters that disproportionately impact older adults. While research has begun to measure disparities, there is a gap in examining wildfire-specific disasters. To address this gap, this scoping review analyzed literature to explore the nexus of wildfires and older adults. We searched peer-reviewed literature using the following inclusion criteria: (1) published in a peer-reviewed journal; (2) available in English; (3) examines at least one topic related to wildfires; and (4) examines how criterion three relates to older adults in at least one way. Authors screened 261 titles and abstracts and 138 were reviewed in full, with 75 articles meeting inclusion criteria. Findings heavily focused on health impacts of wildfires on older adults, particularly of smoke exposure and air quality. While many articles mentioned a need for community-engaged responses that incorporate the needs of older adults, few addressed firsthand experiences of older adults. Other common topics included problems with evacuation, general health impacts, and Indigenous elders’ fire knowledge. Further research is needed at the nexus of wildfires and older adults to highlight both vulnerabilities and needs as well as the unique experience and knowledge of older adults to inform wildfire response strategies and tactics.

## 1. Introduction

The growing threat of climate change has been well-documented in recent years. Since 2011, concentrations of greenhouse gases in the atmosphere have soared, pushing global surface temperatures to an estimated 1.3 degrees Celsius above pre-industrial levels [1]. Human-induced climate change has accelerated impacts of ecological degradation, biodiversity loss, and extreme weather events. These include, but are not limited to, increases in areas burned in wildfires, cyclone intensity attributed to sea-level rise, severe and prolonged droughts, heavier precipitation, and substantial—and in some cases irreversible—damages to biodiversity and ecosystems [1]. These impacts are not felt evenly, with already vulnerable populations suffering the brunt of the crisis. Those living in poverty, women, children, older adults, outdoor workers, people with disabilities, Indigenous populations, and people of color are facing adverse health events. These include increased morbidity and mortality from disease connected to heat stress, exposure to air pollution and smoke, and vector-borne illnesses, in addition to ongoing human rights violations during this era of climate crisis [2].

The social and ecological consequences of wildfires are areas of growing concern, with recent wildfire seasons breaking precedents for frequency and intensity [3]. In the U.S. alone, wildfire events are increasing, with an average of 6.9 million acres burned annually, more than double the annual acreage burned in the 1990s, with the top five worst wildfire seasons in the U.S. all occurring since 2006 [3]. Record-breaking wildfire seasons from Australia to the Arctic and in North and South America are an ominous sign of the ever-growing duration, frequency, and intensity of wildfire seasons to come [4]. Even in the best-case scenarios for curbing emissions, the risk of global wildfire occurrence will still increase by 31–57% by the end of the century [4]. Environmental change related to wildfires is also unique in that wildfires are exacerbated by climate change and are also a contributing factor in the worsening of climate change through the release of greenhouse gasses (GHG) and destruction of carbon stored in trees.

Beyond the environmental impacts, increasing wildfires are also a grave threat to human health. Smoke from wildfires worsens air quality and increases exposure to and inhalation of smoke and small particulates from ash, referred to as particulate matter smaller than 2.5 microns (PM_2.5_) [5,6]. Wildfires lead to increased PM_2.5_ and decreased air quality—increasing the odds of respiratory and health concerns such as burning eyes, runny nose, scratchy throat, headaches, respiratory illness, and exacerbation of pre-existing conditions such as asthma and COPD [6,7]. Breathing wildfire smoke is associated with increased outpatient visits, emergency visits, hospitalization, and death from a myriad of respiratory issues, which is only complicated by the current COVID-19 pandemic, as breathing PM_2.5_ (the primary health concern related to wildfire smoke) is associated with increased morbidity and mortality of the novel coronavirus [7,8].

When a wildfire encroaches upon or destroys communities, emergency preparedness and response and mitigation strategies have also been investigated in conjunction with human health vulnerabilities during times of wildfire disaster. Studies concerning evacuation and emergency service systems in protecting human life and health have been carried out around the world [9]. Many studies indicate significant numbers of people delay evacuation during a wildfire event, often leading to increased evacuation danger [9]. In the immediate aftermath of a wildfire disaster, access to prescription medication, healthcare providers, and mental health services can be lacking [10]. Once these aftershocks have subsided, psychological distress following landscape and ecosystem loss—as well as personal loss or trauma—can be prevalent among the general populations [11].

The very same populations experiencing the most adverse health consequences from climate change are also vulnerable to impacts from other natural disasters, including wildfires, with older adults principal among them. Research has demonstrated that in addition to the usual concerns associated with natural disasters such as injuries and infectious disease outbreaks, older adults face added challenges due to functional or mobility limitations, decreased social supports, difficulty maintaining necessary health regimens, and limited access to information about disaster preparedness and recovery practices [12]. Due to the higher prevalence of chronic conditions among older adults, they often require specialized diets, medicine, and other medical treatments which can be more difficult to maintain or access following the trauma and disruptions caused by natural disasters [13]. Additionally, as people age, their social networks may shrink for a number of reasons, including spouses and close friends passing away or having their children move away, making it more difficult to reach out to others for help [14].

As a result of these age-related risks, older adults are disproportionately negatively impacted by natural disasters when compared to other age groups [15]. For example, while older adults made up only 15% of the New Orleans population, 71% of the people who died from Hurricane Katrina were over the age of 65 [15]. Studies have shown that older adults are often more likely to encounter life-threatening challenges while trying to evacuate during a natural disaster, are less likely to receive disaster warnings, and often experience greater financial losses following the destruction of natural disasters [16]. These disparate outcomes faced by older adults occur with all types of natural disasters, indicating that the needs of this population during these times of crisis need to be addressed [17].

While the disparate impact of natural disasters on older adults is well-documented in scholarly literature, most of this research has focused on hurricanes and flooding [18]. There is a gap in the literature examining the impact of wildfires on older adults [18]. While some findings from other natural disasters (e.g., evacuation, emergency communication, etc.) are relevant across disasters, wildfires have unique health impacts related to smoke and heat exposure that may pose multiple burdens and harms for older adults. This study seeks to examine this gap in the literature through a systematic scoping review of scholarly literature to understand the existing knowledge base on the impact of wildfires on older adults, as well as identify other gaps in data and priorities and directions for interventions and future research.

## 2. Materials and Methods

Due to the lack of literature on wildfires and older adults, the scoping review methodology was chosen due to its usefulness to “determine the scope or coverage of a body of literature on a given topic and give clear indication of the volume of literature and studies available as well as an overview (broad or detailed) of its focus” [19] (p. 2). The scoping review methodological framework followed guidelines from Arksey and O’Malley [20], as well as recommendations by Levac et al. [21] and Cloquhoun et al. [22]. The PRISMA-ScR checklist was followed for documenting and reporting findings [23].

### 2.1. Inclusion Criteria

To answer the research question “What is the extent and scope of literature on wildfires and older adults?” the following inclusion criteria were used:Published in a peer-reviewed journal;Available in English language;Examines at least one topic related to wildfires;Examines how criterion (3) relates to older adults in at least one way.

For criterion (1), peer-reviewed journal publications were chosen to explore academic literature relating to older adults and wildfires to gain an understanding of relevant evidence, themes, needs, and gaps in the literature. For criterion (2), references were limited to the English language due to the research team’s inability to translate articles from other languages. For criterion (3), wildfires were specifically chosen as the disaster of focus due to gaps in the literature exploring the impacts of wildfires (versus other types of disasters such as hurricanes, flooding, etc.) on vulnerable populations, especially older adults. For criterion (4), we defined older adults as those who are 60 years and older, or who were referred to in the references as “older adults”, “elderly”, “elders”, etc. (see search terms below). This age was chosen based on literature indicating that 60 is a common parameter for identifying this age group [24]. Further, criterion (4) means that included articles must specifically connect wildfires to older adults in some way, excluding those that discussed older adults and wildfires separately.

### 2.2. Literature Search and Screening

Search terms and protocols were established in consultation with a university librarian. Based on these discussions, the following databases were searched: PubMed, Web of Science, ProQuest (Agriculture and Environmental Sciences Collection, Sociological Abstracts, and Social Service Abstracts), and EBSCO Host (Academic Search Complete, Environment Complete, GreenFILE, PsycInfo, and SocINDEX).

In consultation with the librarian, the following search strings were created and run in each database:“older adult*” OR senior* OR elder* OR “older person*” OR “older people” OR geriatric* OR gerontolog* OR “old age” OR “long term care” OR “nursing home*” OR “assisted living” OR “independent living” OR “skilled nursing facilit*” OR “memory care” OR “residential care” OR “retirement communit*”;

AND2.wildfire* OR “wild fire*” OR bushfire* OR “bush fire*” OR bushfire* OR “forest fire*” OR “brush fire*” OR brushfire* OR “wildland fire*” OR “uncontrolled fire*” OR “fire season*”.

Both search strings were searched “anywhere but full text (NOFT)” within the ProQuest database, and with the default search settings for other databases, which was the recommendation and guidance of the university librarian. The search, conducted in March of 2021, yielded 585 articles. After removing duplicate records, 261 remained.

The research team used Covidence systematic review software [25] to complete the screening process. Two authors independently reviewed the titles and abstracts of the 261 non-duplicate records. After title and abstract screening, 138 remained. Two authors then independently read the full text of these 138 remaining articles. Of these, 75 met the inclusion criteria (Figure 1) [18,26,27,28,29,30,31,32,33,34,35,36,37,38,39,40,41,42,43,44,45,46,47,48,49,50,51,52,53,54,55,56,57,58,59,60,61,62,63,64,65,66,67,68,69,70,71,72,73,74,75,76,77,78,79,80,81,82,83,84,85,86,87,88,89,90,91,92,93,94,95,96,97,98,99]. Throughout the screening and review process, any disagreements on inclusion/exclusion were discussed and reconciled as a team before making a final decision.

### 2.3. Data Extraction and Analysis

Data collected on each article included: (1) article characteristics and type; (2) information related to environmental issues including the disaster recovery cycle, specific hazards, etc.; (3) information on how older adults were included and relevant findings; and (4) whether articles addressed problems, used responses or interventions, or suggested solutions, recommendations, or areas of future research. We created, pilot tested, and refined our data collection tool using Google Forms. Once the final form was created, two members of the research team independently recorded data from each article. Any questions or disagreements were discussed and resolved as a team. During analysis, we also identified thematic topics arising from the literature.

First, basic characteristics included the year of publication, article title, author(s), journal title, country or geographic focus, study type, sample, and methods used. Second, information related to environmental issues included the hazards addressed (wildfires, air quality, heat, haze, or other types of hazards); specific disasters addressed; explicitly mentioning climate change or recommendations for climate adaptation and/or mitigation; focus on any part of the disaster recovery cycle (response, recovery, mitigation, preparedness); explicitly mentioning environmental justice or alluding to it; and the inclusion of Indigenous or Aboriginal traditional ecological knowledge (TEK). Third, questions related to older adults included whether the primary focus of the article was older adults and/or how older adults were included; focus on older adults in the community or in residential facilities; and relevant findings or recommendations related to older adults. Fourth, questions related to study focus included focus on problem description, measuring exacerbation of specific health problems, inclusion of responses or interventions, inclusion of Indigenous or Aboriginal knowledge of fire management, solutions or recommendations, areas of future research, and thematic topics arising in the literature. All criterion, except for thematic topics, were established during the creation and pilot testing of the data collection tool. Thematic topics arose during data collection as patterns emerged in the literature.

## 3. Results

### 3.1. Basic Characteristics of the Literature

A total of 75 peer-reviewed journal articles met study inclusion criteria. There was no limit on year of publication in our initial search; the earliest article was published in 2001, with the frequency of publications increasing over time (Figure 2). Only 3 of the 75 articles (4%) were published between 2001 and 2006, 10 (13.3%) were published between 2007 and 2011, 23 (30.7%) were published between 2012 and 2016, and 39 (52%) were published between 2017 and 2021 (Figure 2).

Geographic regions discussed were diverse, but the majority were based in North America (44%) and Oceania and Australia (26.7%), and many were about wildfires or fire management on First Nations or Tribal land (18.7%) (Figure 3). The United States (U.S.) was the most represented country, representing 26 of the 33 total mentions of North America. Most of these focused on the western U.S. (*n* = 12), specifically California (*n* = 8). Of the eight articles focused on Canada, six were about wildfires on First Nations land. The 20 articles focusing on Oceania and Australia were almost exclusively focused on Australia, with 1 mentioning New Zealand and 6 of the 20 focusing on Aboriginal land. Seven articles focused on Northern and Western Europe (Spain, Portugal, Greece, and two from Finland), and five articles focused on South America, all of which were in Brazil’s Amazon region. Five articles were based in Southeastern Asia (two in Malaysia, two in Indonesia, and one covering Singapore, Malaysia, Indonesia, Brunei, and Thailand). Finally, those that covered more than three countries were labeled as “global”, though these predominantly focused on countries above including the U.S., Australia, Malaysia, and Indonesia. Notably, no articles covered geographic regions of Africa, Central America, or North and Central Asia, though one global article mentioned “Asia, Latin America, and Africa” [42] (p. 99).

Of the 75 articles, 63 (84%) were empirical research articles or evaluations (Table 1). Most of these were quantitative (44%) or qualitative (26.7%), with a few being mixed methods (4%) or systematic reviews (9.3%). The 12 non-empirical articles (16%) were conceptual, descriptive, or commentaries. Methods used in empirical articles varied, with secondary data (37.3%) being the most prevalent. A large majority of articles focused on measuring morbidity and mortality related to wildfire smoke, with 22 articles (29.3%) using emergency room and hospital admissions or mortality rates as secondary data. The second most common method was remote-sensed environmental measures (29.3%), measuring air quality and pollution, particularly of PM_2.5_ levels and other particulate matter. Interviewing was the third most prevalent method (21.3%). Other methods included systematic reviews (9.3%), surveys (9.3%), focus groups (8%), case studies (6.7%), field research (5.3%), biological data (5.3%), and other methods (8%; e.g., participatory action research, future modeling, ethnography, and Q methodology) (Table 1).

### 3.2. Environmental: Hazards, Climate Change, and Disaster Recovery Cycle

We reviewed articles for specific information related to environmental issues including specific wildfires, other hazards, and language or information about climate change, environmental justice, or the disaster recovery cycle (Table 2). All articles discussed wildfires, bushfires, or forest fires in some way. Some articles also discussed other types of disasters such as flooding and hurricanes, but due to the proliferation of literature on these topics, we only collected data on hazards related to wildfires. Of these related hazards, 41 articles discussed air quality (54.7%), 12 covered heat (16%), and five discussed haze (6.7%). Almost half of articles were either about a specific wildfire (17.3%) or a specified wildfire season or time period where wildfires occurred (25.3%). Wildfire events or seasons that were covered in more than one article included: wildfires and associated “haze disaster” in Indonesia in 1997 [49,53]; wildfires in San Diego, California in 2007 [31,33]; a 2011 wildfire impacting Sandy Lake First Nation in Canada [27,28]; California’s 2017–2018 wildfire season [43,47,97]; and the catastrophic 2019–2020 wildfire season in southeastern Australia [40,50,89].

Data have consistently shown that climate change is increasing the intensity and impact of wildfires [1,18,60]. However, not all disaster research makes the connection between climate change-related causes and impacts. We found that 41 articles (54.7%) mentioned climate change or global warming explicitly, but only 7 (9.3%) focused on climate change as a main topic. We also collected data on interventions, recommendations, or responses that may be climate mitigation or adaptation strategies, even if they were not named as such. We found that 41 articles (54.7%) addressed some form of adaptation strategies and 18 (24%) addressed mitigation strategies (Table 2).

In addition to climate change, data were collected on mentions of the disaster recovery cycle and specific phases including recovery, response, mitigation, and preparation (Table 2). A majority of articles mentioned the disaster recovery cycle or at least one phase (56%). Mitigation measures were the most prevalent phase discussed (33.3%), closely followed by response (32%) and preparation (29.3%). Recovery was the least discussed phase, addressed by eight articles (10.7%).

Because of the particular vulnerability of older adults to disasters, including wildfires, we noted whether articles specifically mentioned environmental justice. However, during analysis, we found that many articles alluded to environmental justice by discussing “disadvantaged and vulnerable populations” [65] or “vulnerable populations, including the elderly, socioeconomically disadvantaged groups, and those with underlying chronic disease… [who are] most affected [29]. While only two (3%) articles explicitly named environmental justice [46,60,64], more than half (53%) alluded to environmental justice by discussing disproportionate impacts or particularly vulnerable populations in some way (Table 2).

### 3.3. Older Adult Findings

When reviewing how articles discussed older adults (Table 3), 39 articles defined older adults based on either an age cutoff (e.g., 65 or older) or by naming this population (e.g., elders, older adults, seniors, etc.). A large portion of articles (41.3%) included older adults as a population they were specifically interested in looking at in addition to others, while 29.3% focused solely on older adults, and the remaining 29.3% made mention of this age group but did not have them as their primary focus. The majority of articles (69.3%) based their findings on older adults using information that was collected about them, rather than from them firsthand (24%), with some (6.7%) doing both. Most articles did not explicitly state the living conditions of the older adults that were included; however, out of the 24 articles that did make this specification, 20 focused on older adults living in the community while only 4 focused on older adults living in long-term care communities.

With respect to the findings and recommendations made for older adults in the context of wildfires, articles discussed the various ways that older adults are impacted by and respond to wildfires. A majority of articles (60%) discussed the health impacts that wildfires had on older adults, describing increased hospitalization and death rates for cardiovascular and respiratory issues during or following wildfires for this population [26,53,66,82,90]. These negative outcomes were increasingly worse for older women and older adults of color [60]. While a meta-analysis of these impact estimates was beyond the scope of this study, some examples of specific findings include a 7.2% increase in respiratory hospital admissions among Medicare enrollees in the Western U.S. during intense smoke days [59] and impairments to lung function, especially among the elderly, of 33.9% of participants at two-years post-exposure to smoke from a Montana wildfire in the U.S. [78].

Additionally, when it came to responding to a wildfire, most notably with evacuations, older adults faced a disproportionate amount of barriers and challenges including difficulty maintaining the level of care they needed, accessing medications, and staying connected with caregivers, demonstrating how the needs of older adults may not be fully considered and addressed during wildfire disasters [27,28]. Finally, findings illustrated the role that older adults play during wildfires in supporting their local community, family members, and friends. During evacuations, older adults offered additional support to one another by making meals for one another, helping with laundry, and providing emotional support [27,28].

### 3.4. Thematic Topics, Problem-Focus, Interventions, Recommendations, and Future Research

While reviewing included articles, authors made note of recurring themes of interest that provided additional insight on the impacts and experiences felt by older adults due to wildfires (Table 4). With respect to the experiences of older adults, 17.3% (*n =* 13) of articles discussed animals/pets, 12% (*n* = 9) included caregivers, 34.7% (*n* = 26) touched on evacuation efforts and experiences, 14.7% (*n* = 11), focused on intergenerational relationships during wildfires, and 37.3% (*n* = 28) mentioned the effect of social support/social capital for this population during these disasters. Additionally, some articles discussed more specific impacts on older adults during wildfires, including 25.3% (*n* = 19) that looked at mental health associations, and 48% (*n* = 36) focused on morbidity and/or mortality of wildfires and associated hazards (air pollution, particulate matter, heat, etc.) from an epidemiological focus on population health. Finally, it should be noted that the onset of the COVID-19 pandemic brought about additional issues, especially as they relate to an older adult’s health and well-being, and 2.7% (*n* = 2) of articles discussed the added complexity to the impact of wildfires.

Of the 75 articles, 56 (74.4%) were problem-focused, describing negative impacts of wildfires in some way (e.g., need for evacuation, impacts of air quality, needs of communities, etc.) (Table 5). Of the 56 that focused on problem description, 36 (48%) described problems of morbidity or mortality related to wildfires and/or wildfire smoke. Most of these used secondary, epidemiologic data such as hospital admissions and death rates to describe the health impacts of wildfire smoke and/or PM_2.5_ (particulate matter smaller than 2.5 microns). Aside from morbidity and mortality, other articles described problems with evacuation, displacement, and/or issues with disaster response [27,28,40,47,86,87,91,96,97]; inequalities and vulnerabilities of certain populations to wildfires [29,77,79]; and general descriptions of health impacts without epidemiologic data [35,42,80].

Many articles moved beyond problem description with 32 of the 75 (41.3%) articles describing responses or interventions during, after, or in preparation for wildfires. Interventions and responses included individual, organizational, and community-level efforts. Individual efforts included masking to avoid smoke exposure [53,84], installing in-home air filters [34,41,84], and creating survival plans [76,91,96]. Organizational interventions predominantly focused on organizations (e.g., long-term care facilities, rehabs, and hospitals) evacuation and/or disaster management plans [27,28,30,31,33,52,86,97], but also included treatment recommendations for providers [42,92]. Community-level responses included descriptions of families and neighbors caring for one another during acute disaster phases [27,28,30,40,50], and disaster management and coordinating systems at the community level [28,30,63,87]. Finally, many articles described traditional ecological knowledge (TEK) or Indigenous and Aboriginal elders as an important intervention for “hazard abatement” [55], as well as the opportunity for fire management institutions to listen to, learn from, and rematriate (e.g., return power to Indigenous peoples to reclaim ancestral traditions) [100] fire “management“ as well as the ethics of fire management agencies “using” this knowledge [26,38,64,67,68,69,70,73,75,94,98].

In addition to interventions and solutions, 61 (81.3%) articles provided recommendations targeted at multiple levels and points of intervention including individuals, organizations, communities, policy, scholarly literature, and disaster and fire management agencies. Many recommendations intersected with other findings, such as recommended adaptation strategies [29,58,61,66,71,88] and the importance of individual survival plans, community evacuation plans, and organizational disaster management protocols and plans, especially in relation to communicating with older adults [28,39,52,54,76,77,79,81,97]. Community-centered disaster management planning and strategies were prolific across recommendations, with 26 of the 31 (83%) discussing community needs, community engagement, or community inclusion in disaster management planning in some way (Table 5).

Finally, most articles (81.3%) outlined areas for future research, describing the importance of utilizing more rigorous and longitudinal research methods to examine the long-term health effects on older adults due to wildfires, especially those from more minoritized communities (Table 5). Additionally, findings suggest community and local government officials need to consider the needs of older adults during wildfires and research should serve as a tool to evaluate the short- and long-term impacts of responses and interventions through all phases of the disaster recovery cycle [2,7].

### 3.5. Study Strengths and Limitations

One strength of this review is its systematic and rigorous approach to identifying relevant peer-reviewed literature, by using expansive search terms and searching more than 10 databases. This allowed a breadth of literature to be explored across geographic regions, fields of study, and disciplines. However, one limitation is the exclusion of gray literature (e.g., books, non-peer-reviewed articles, etc.) that may have had additional information related to the impact of wildfires on older adults and relevant recommendations or interventions. Further, our search was limited to publications available in English, which excluded two potential studies from full review, as well as other non-English publications that may have been excluded from our initial database search.

## 4. Discussion

### 4.1. Wildfires and Older Adults: Increased Engagement and Trends

While there is prolific literature on the impact of extreme heat and hurricanes on older adults, there is a gap in the literature “on the vulnerability of older adults to other health-related climate impacts, such as…wildfire [and] changes in air quality” [18] (p. 21). This review systematically synthesized scholarly literature focusing on older adults and wildfires to help identify priorities and directions for addressing gaps in the literature on the impact of wildfires on older adults, and recommendations for interventions and future research. In a global search with no restriction on publication date, only 75 articles were found and most (52%) were published within the past 5 years (2017–2021). This may indicate the impact of wildfires on older adults is a newer area of research that requires additional exploration and evaluation.

Wildfires may have unique health impacts that spread beyond a specific boundary where the disaster occurred, as smoke and air quality transcend boundaries, with smoke from large fires sometimes traveling thousands of miles, across countries and even continents [54,101,102]. This was seen in multiple articles, with some specifically addressing “long-range transboundary air pollution” [54,85] and others examining health-related impacts of air quality even when the source of the fire was in a different geographic location [53,82,85].

Findings from this review show the particular vulnerabilities of older adults to wildfires, particularly due to poor air quality and exposure to smoke and particulate matter (i.e., PM_2.5_). Many articles within the review explained that older adults are more susceptible to adverse health impacts of PM_2.5_ [29,34,37,42], as are those with pre-existing respiratory or cardiovascular diseases and those with lower socioeconomic status (SES) [29,34,72]. While older adults are named as specifically susceptible, they also often have pre-existing conditions or may have lower incomes, exhibiting a double—or triple—burden related to poor air quality. While there is substantial research on health impacts related to particulate matter, some studies have found that PM_2.5_ exposure from wildfires may be more toxic than equal doses of ambient PM_2.5_ [59,103], highlighting the importance of examining wildfire-related air quality and health impacts, especially for older adults.

Impacts of air quality are compounded by heat exposure—another hazard related to wildfires. Many articles spoke to the health impacts of heat on older adults particularly, highlighting “the double burden that heat and socioeconomics play for low-income older adults who are unable to afford air conditioning or caregiver support during extreme heat” [66] (p. 7). Heat-related deaths are the most deadly “natural disaster”, and accompany wildfires—along with poor air quality—illustrating the impact of wildfires on older adults even if they are not directly exposed to the epicenter of a wildfire event [36].

Aside from indirect—albeit very real—impacts of wildfires through air quality and smoke, many articles discussed acute phases of the disaster recovery cycle when a wildfire occurs, namely the response phase (32%) and evacuation (34.7%). The findings showed that older adults are particularly vulnerable during evacuation phases, noting the importance of considering elders when planning for community-level communications for evacuation [47,79,81] and physical difficulties elders may have with evacuation, especially without social support [18,27,28,79]. Even if older adults are not evacuated, being in the geographic region of a wildfire event with power outages may affect life-sustaining equipment such as oxygen, ventilators, CPAP machines, refrigeration for medications, power wheelchairs, elevators, and heating and cooling systems for body regulation [18,79]. Wildfires may also pose a threat to the continuity of care for older adults who need ongoing medical treatment such as dialysis, cancer treatment, obtaining medications, or other medical needs [18,79,97].

### 4.2. Dominant Narratives: Secondary Data and Epidemiological Studies

The most prolific finding in this review was the use of secondary data to measure morbidity and mortality from wildfires or associated hazards (e.g., heat, air quality, etc.). This aligns with findings from an included article stating, “in relation to extreme weather conditions, literature has highlighted the vulnerability of older adults as a cohort, though there is limited attention on how to prevent the cohort from experiencing increased risk” [39] (p. 974). The majority of articles (74.7%) focused on problem description, with 48% of all articles describing the problem of morbidity and mortality impacts—either focusing on older adults or whose findings skewed towards older adults. This illustrates the dominant narrative of wildfires and older adults, telling a story of risk and vulnerability. While many articles also discussed responses or interventions, these were still predominantly focused on describing problems within the intervention or response itself, such as lessons learned from evacuation or community responses or preparation. Epidemiologic findings are imperative to provide statistics to build a base of scientific knowledge about this issue, but they only tell a fraction of the story about older adults, leaving out vital information from older adults on their lived experiences and needs before, during, and after wildfires.

### 4.3. Older Adults: Lived Experiences and Primary Data Sources

The results demonstrate how most of the information on the intersection of wildfires and older adults is primarily data collected about older adults from other sources rather than from this population firsthand. Medical and hospital records were one of the main sources of information that articles drew from, focusing on the negative physical health effects of wildfires on this population, but articles rarely focused on learning from what older adults went through or how they felt about wildfires and their role in relation to these disasters. To adequately address the disproportionately negative issues faced by older adults in the face of wildfires, it is essential to better understand their perspectives and what they find to be their greatest challenges and needs during these disasters. Articles also demonstrated how older adults can be a vital source of knowledge in knowing how to reduce or respond to wildfires, as evidenced by the numerous articles on the role Indigenous elders have previously had in mitigation efforts (see also Section 4.4) [38,55,64,67,68,69,70,73,74,83,94,98]. It is important to understand that older adults are not simply victims of wildfires but can, in fact, play a major role in addressing these growing disasters.

### 4.4. Social Support and Community Focus

Articles demonstrated the importance of social support for older adults at both a community and individual level. Older adults who lacked social support were more likely to die during a wildfire as they did not receive adequate warning of the danger or were unable to evacuate on their own [43,79]. Caregivers were noted as a vital source of support for older adults but were still in need of the appropriate resources and financial assistance to prepare for and respond to wildfires [97]. Caregivers should be considered a valuable point of contact for older adults in providing needed public health and disaster response messaging to this population [81]. When formal institutional responses were not adequate in meeting the needs of older adults, articles stressed the importance and power of informal neighborhood and community responses to make up for this lack of support [43,50]. In fact, one study found the number of fatalities due to wildfires was reduced when communities supported their older adults [43]. A good social support network was also found to provide critical psychological and emotional support for older adults during wildfire evacuations, which older adults cited as the most prevalent and valuable support they received during this crisis [27,28].

Building on the importance of social support and community care, many articles discussed the need for community-engaged tactics within disaster management systems including first responders and emergency management agencies. Articles discussed the need for community-responsive practices, with one article asserting “community engagement to determine most appropriate strategies from the local level should become a focus of adaptation. For example, bushfire preparedness and management should incorporate knowledge of community, government, and industry groups to identify impacts on community safety” [29] (p. 754). Other articles reiterated this, highlighting the need to build partnerships between local, state, and federal emergency management and public health systems, and that these should be in conversation and relationship with community members and responsive to their needs [32,35]. Other findings highlighted the need for communication strategies to be developed in conjunction with communities [34,35] and a need for more education and “community activism…to promote outreach that assists vulnerable persons [e.g., older adults] during emerging hazardous weather situations” [43] (p. 383).

### 4.5. Elders and Traditional Ecological Knowledge (TEK)

An important finding of this review was the inclusion of Indigenous, First Nations, and/or Aboriginal elders’ experiences and knowledge of fire. Almost 20% (*n* = 14) of articles focused explicitly on Indigenous, First Nations, or Aboriginal elders, with 12 focusing on fire knowledge and TEK and 2 focusing on the impacts of evacuation during a fire event [27,28]. While it is beyond the scope of this paper to fully explore the relevance of TEK to wildfires, this emergent finding became salient during data collection and analysis due to the volume of related articles. These articles highlighted the importance of community in a different way, illustrating the deeply held community and cultural ties of Indigenous peoples to each other and the land. In contrast with other articles focusing on evacuation, those focused on the evacuation of Indigenous peoples highlighted a deeper sense of social cohesion and therefore social disruption when evacuations occurred. A participant from one article discussed the way evacuation broke up “communityness” stating, “the evacuation breaks up families, it breaks up that ‘communityness’, how you feel home. It breaks that up and you’re being sent to a strange land” [27] (p. 372). These findings illustrate not only recommendations for Indigenous elders during evacuations, but also aspects of building “communityness” and social cohesion that other communities may learn from as a form of disaster preparation and response.

The majority of articles focused on Indigenous elders’ fire knowledge and how this contrasts with dominant “fire management” agencies, policies, and protocols. Fire knowledge included cultural and traditional burning practices that have been utilized by Indigenous peoples for generations, and how fire knowledge is a part of sacred and cultural practices of being in relationship with the land. Almost all of these articles discussed implications of fire knowledge for fire management agencies, and many included cross-cultural dialogues or comparisons between Indigenous elders and other fire management agencies [38,55,64,67,68,69,75,83]. Many of these articles discussed the difference between Indigenous peoples’ ontological views of fire and those of fire management institutions, most of which are run by White settler nations (e.g., Australia, Canada, and the United States). One article explained the difference between TEK and scientific ecological knowledge (SEK) [64], explaining that TEK takes a relational view of nature whereas SEK views nature through lenses of control, domination, and subjugation. Other articles affirmed this, explaining the incongruity of “fire management” or “fire-fighting” with TEK’s view of fire and land as something to be in balance and relationship with rather than managed or fought [38,64,68,75].

Findings provided examples for collaborative co-management between Indigenous elders and fire management agencies, highlighting the importance and potential of TEK in fire “management” practices, while also naming the tension and ethics of non-Indigenous peoples “using” TEK for fire mitigation and/or adaptation measures [64,70]. One article explained that Indigenous elders have difficulty trusting fire management agencies run by the government due to generational trauma of genocide, relocation, and colonialism, with an Indigenous elder stating “science means not us” [83] (p. 26). Other articles provided recommendations for adaptive co-management strategies to build relationships between fire management agencies and Indigenous peoples to create “cross-cultural partnerships directed towards fostering resilience” [68,69,70]. These findings illustrate a nuanced and complex picture of the role of TEK in fire “management”. Indigenous people have been care-takers of the land for generations and TEK must be incorporated into any understanding of ecological care, including wildfire management. While ethics and use of TEK are beyond the scope of this paper, there is a breadth of literature that looks at the intersection of TEK and fire “management”, building on the articles related to TEK in this review. For the purpose of this paper, these findings illustrate not only the impact that wildfires have on older adults, but also the positive impact older adults can have on adaptation, mitigation, or responses to wildfires.

### 4.6. Increased Focus on (Un)Natural Disasters: Climate Change and Environmental Justice

While a primary focus of this paper was the impact of wildfires on older adults, this impact cannot be understood without analyzing the causes of wildfires. While wildfires are not new, the frequency and intensity of wildfires have dramatically increased due to climate change, creating (un)natural disasters [1]. While only 7 (9.3%) articles had a primary focus on climate change, 41 (51.7%) mentioned climate change as a reason for increasing disasters, reaffirming the relationship between worsening wildfires and climate change. Of the seven articles focused on climate change, three highlighted the disproportionate impact of climate change on older adults [18,66,71] while others had findings that skewed towards older adults [29,72,80,88]. However, the low number of articles focusing on the intersection of climate change and older adults illustrates a need for further research in this area, particularly the relationship between climate change, wildfires, and older adults. This intersection will only become more pertinent, as 8 of the 10 worst global wildfire seasons have happened in the past decade. Coupling the increased intensity and frequency of wildfires with the ongoing COVID-19 pandemic, further research is needed to examine these intersectional crises and their impacts on older adults.

The findings from this review continually reiterated that the impact of disasters is not distributed equally. The disproportionate impact of environmental hazards on some groups of people more than others is known as environmental injustice. While only 2 (3%) articles named environmental justice specifically, 40 (53%) of articles alluded to environmental justice in some way. Most of these references were related to older adults as being particularly “vulnerable” to wildfires and associated hazards (e.g., heat, air quality), while others provided a more nuanced understanding of environmental justice with other intersecting identities such as race, class, ethnicity, gender, geographic location (i.e., urban versus rural), and socioeconomic status [18,29,64,65,66,77]. While environmental justice is well-documented within scholarly literature, these findings point to the importance of incorporating an environmental justice perspective into research on wildfires and older adults. Some articles that used the “dominant narrative” named above (i.e., epidemiological studies of morbidity and mortality) used variables to understand the impacts of intersectional identities, providing a framework to incorporate environmental justice, though others named the gap in understanding environmental justice through public health-related data [65]. Some articles explicitly named this as a limitation or need for future research [34,47,49,57,65]. Future studies of all kinds should incorporate environmental justice into their data collection, methods, or analysis to understand the nuanced and disproportionate burden or wildfires on vulnerable populations and ways to address these harms and uneven impacts. Studies may also build upon this literature base by incorporating climate justice into environmental justice, especially in the case of increasing and worsening wildfires [104].

## 5. Conclusions

Findings from this scoping review demonstrate how older adults can be an important source of knowledge for wildfire mitigation, response, recovery, and adaptation strategies and should be included in local community planning efforts. Additional efforts should be made to incorporate environmental justice and intersectionality to better understand the root causes in health disparities among older adults during and following wildfires. Overall, the literature on the different ways older adults respond to or are impacted by wildfires is still relatively new and needs further development and exploration to better learn from and support this population in the face of worsening wildfire disasters.

## Figures and Tables

**Figure 1 ijerph-20-06252-f001:**
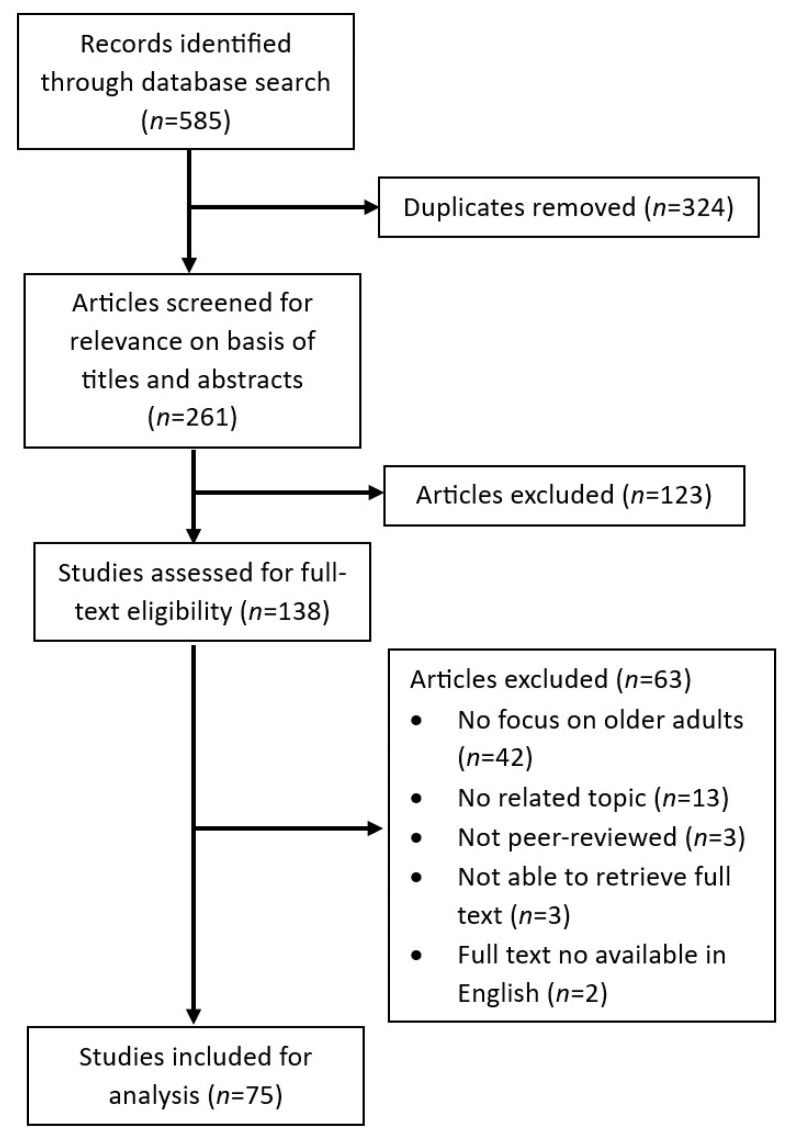
Flowchart of search, screen, and review process.

**Figure 2 ijerph-20-06252-f002:**
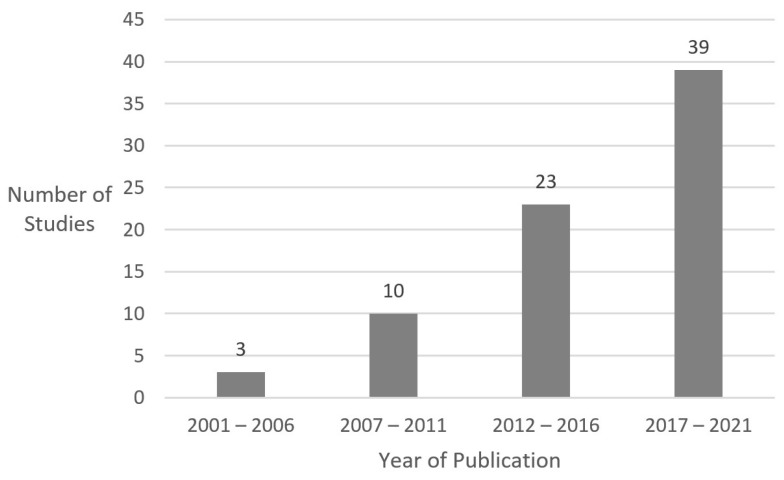
Year of publication.

**Figure 3 ijerph-20-06252-f003:**
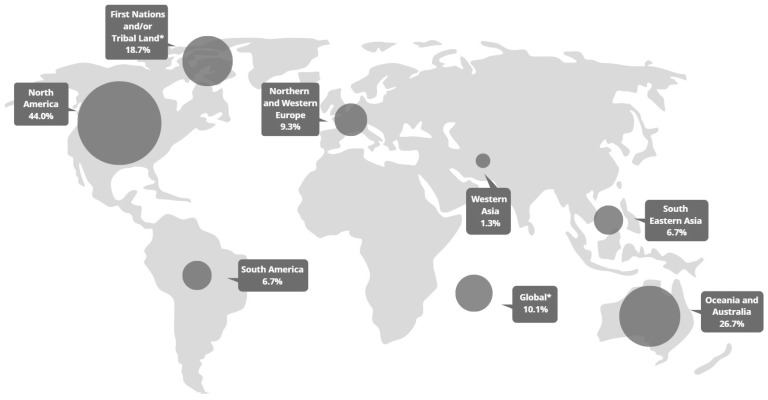
Geographic focus of articles. Note: Percentage exceeds 100% as some articles covered multiple regions. * Articles focusing on First Nations and/or Tribal lands were based in North America (Canada, *n* = 7; United States, *n* = 3) and Oceania and Australia (Australia, *n* = 6).

**Table 1 ijerph-20-06252-t001:** Basic characteristics of the literature (*n* = 75).

Characteristic	*n* (%)
Paper Type	
Quantitative	33 (44)
Qualitative	20 (26.7)
Mixed Methods	3 (4)
Systematic/Scoping Review	7 (9.3)
Conceptual Papers	9 (12)
Other (e.g., commentary, interview transcript)	3 (4)
Method	
Secondary Data	28 (37.3)
Remote-Sensed Environmental Measures (Air Quality)	22 (29.3)
Interviews	16 (21.3)
Systematic Review	7 (9.3)
Survey	7 (9.3)
Focus Group	6 (8)
Case Study	5 (6.7)
Field Research	4 (5.3)
Biological Data	4 (5.3)
Other Methods	6 (8)
Not Applicable (e.g., conceptual papers or other paper type)	12 (16)

Note: Methods percentage exceeds 100% as some articles used multiple methods and/or data collection strategies.

**Table 2 ijerph-20-06252-t002:** Environmental hazards, climate change, and disaster recovery cycle (*N* = 75).

Environmental Categories	*n* (%)	Examples
Hazards		
Fire	75 (100)	See articles with asterisks in reference list [18,26,27,28,29,30,31,32,33,34,35,36,37,38,39,40,41,42,43,44,45,46,47,48,49,50,51,52,53,54,55,56,57,58,59,60,61,62,63,64,65,66,67,68,69,70,71,72,73,74,75,76,77,78,79,80,81,82,83,84,85,86,87,88,89,90,91,92,93,94,95,96,97,98,99]
Air Quality	41 (54.7)	Impacts of air quality and/or particulate matter [26,29,32,34,37,41,42,44,45,46,47,48,49,51,54,56,57,58,59,60,61,62,65,66,71,72,74,78,88,89,90]
Heat	12 (16)	Impacts of heat [18,29,36,39,40,43,57,66,72,80,87,88]
Haze	5 (6.7)	Haze disasters and/or impacts of haze [48,53,54,82,85]
Specific Wildfire(s)		
Specific Wildfire	13 (17.3)	1997 wildfire and “haze disaster” in Indonesia [49,53] 2007 wildfire in San Diego, CA [31,33] 2011 wildfire in Canada impacting Sandy Lake First Nation [27,28]
Wildfire Season/ Time Period	19 (25.3)	California’s 2017–2018 wildfire season [43,47,97] Australia’s 2019–2020 wildfire season [40,50,89]
Climate Change (CC)		
Mentions	41 (54.7)	
Adaptation	41 (54.7)	Individual-focused adaptations (e.g., adapting to heat, addressing disease burden, air filtration systems, individual survival plans) [18,29,34,39,41,42,61,66,80,84,85,88,91,96] Facility or community-level emergency protocols (planning, preparation, evacuation, communication, etc.) [18,26,27,29,33,47,52,63,79,81,87,97] Land-use management (including traditional ecological knowledge and burning practices) [29,36,38,50,55,67,68,69,70,75,83,94,98]
Mitigation	18 (24)	Traditional ecological knowledge and burning practices [38,55,64,67,68,69,70,73,74,83,94,98] Mentions or addresses need to reduce greenhouse gas emissions [29,58,71,80,88,93]
Article Focuses on CC	7 (9.3)	Health impacts of climate change [29,72,80,88] Disproportionate impact on older adults [18,66,71]
Disaster Recovery Cycle		
Mentions	42 (56)	
Recovery	9 (12)	Needs of older adults in recovery period following wildfires (e.g., disruption in continuity of care, physical recovery, economic recovery, and trauma/mental health) [29,32,79,81,97] Community recovery [50] Debriefing sessions with facility staff following wildfire [31,86]
Response	24 (32)	Needs of older adults during acute wildfire disaster (e.g., life-support equipment such as oxygen during power outages, immediate interventions for air quality, etc.) [9,17,18,34,79] Evacuation (individuals, facilities, communities) [27,28,30,31,40,47,52,76,86,87,91,96] Early warning systems, communication, and local response protocols [18,29,33,34,63,96] Social support needs (families, caregivers, etc.) [18,27,28,29,79,81] Response of health care providers and/or facilities [32,42,52,66,86,92,97]
Mitigation	25 (33.3)	Building codes and updates, and facility emergency protocols [18,29,71] Mitigating smoke exposure [34,41,88] Public outreach, local contingency planning, community risk-reduction etc. [47,63,77,81,87] Reintroducing “ecologically beneficial fire” [35] (p. 677) and Indigenous burning practices [35,38,55,64,67,68,69,70,73,83,94,98]
Preparation	22 (29.3)	Barriers or facilitators to preparedness for older adults (e.g., socioeconomic factors, mobility and health issues, etc.) [39,77,79] Incorporating needs of older adults into planning measures (recommendations, community-engagement, etc.) [29,32,63,66,79,87] Recommendations for evacuation preparedness and/or facility emergency protocols [27,28,33,52,86,98] Individual preparation (survival plans, preparing personal property, evacuating, etc.) [40,76,91,97]
Environmental Justice (EJ)		
Explicit mention of EJ	2 (3)	Intersectional analysis of subgroups of older adults most impacted by wildfire smoke using an environmental justice lens (e.g., race, gender, education) [40] Mention of environmental justice as factor of vulnerability for respiratory disease [26]
Alludes to EJ	40 (53)	Intersectional view of impacted older adults (more impacted based on race, socioeconomic status, gender, housing status, chronic disease, urban vs. rural, and/or) [18,29,32,34,36,39,64,65,66,77] Calls for more focus on vulnerable populations in future research [18,34,57,65,71,82] Connection of Indigenous sovereignty and knowledge, colonization, historical oppression, and resistance [38,55,64,69,70,75,83,94]
No Mention of EJ	33 (44)	
Indigenous or Aboriginal Peoples		
Traditional Fire Knowledge	12 (16)	Co-management strategies and tensions between Indigenous elders and peoples and state, national, or other fire management groups [38,64,67,68,69,70,83] Western science’s need for Indigenous knowledge and tension between the two [38,64,75,94] Description of Indigenous fire knowledge, experiences, and/or history [28,38,55,73,75,94,98]
Focus on Indigenous or Aboriginal Lands	14 (18.7)	In addition to the above 12 articles, 2 focused on community experiences and needs during evacuation for Sand Lake First Nation [27,28]

**Table 3 ijerph-20-06252-t003:** Older adult findings (*N* = 75).

Categories Related to Older Adults	*n* (%)	Examples
Focus Demographic		
Older Adults Sole Focus	22 (29.3)	Focus on Indigenous elders [28,68,69,70] Focus on health impacts of older adults in disasters [18,32,54,60]
Focus on Older Adults in Addition to Others	31 (41.3)	Focus on Indigenous elders in addition to others (other Indigenous people, non-Indigenous land management decision makers, etc.) [27,29,38] Participants were stratified by age or age groups and included both older adults and younger participants [34,65,71,88,90,99]
Mentioned Older Adults, but not Focus	22 (29.3)	Mentioned older adults as another group that could be affected but was not specifically studied in the article [35,47,63,76]
Data Sources		
From Older Adults	18 (24)	Older adults participated in the study (e.g., completed survey, participated in interview, etc.) [70,75,83,91,94]
About Older Adults	52 (69.3)	Medical records about older adults were obtained and analyzed [26,32,48,72,74] Other individuals contributed information about older adults (e.g., caregivers, health professionals, first responders, etc.) [52,63]
Both	5 (6.7)	A combination of information shared by older adults and obtained about older adults was used concurrently in the study [27,28,35,40,78]
Living Environment		
Community	20 (26.7)	Articles focus on older adults living in community, not in long-term care [27,40,41,55,97]
Long-Term Care	4 (5.3)	Articles focus on older adults living in long-term care communities [31,33,66,86]
Not Specified	51 (68)	Articles do not specify the living setting of the older adult[s] in the study [26,44,56,62,78,95]

**Table 4 ijerph-20-06252-t004:** Specific topics and themes (*N* = 75).

Thematic Topic	*n* (%)	Examples
Animals/Pets	13 (17.3)	Traditional ecological knowledge including importance of animals in landscape, ecosystem, or relationality between humans and the more-than-human world [55,67,68,69,70,75,83,94,98] “Animal guardians” or “animal ownership” and its impact on evacuation, preparedness, and/or emergency response [76,91,96]
Caregivers	9 (12)	Importance of having caregivers of older adults involved in and/or educated on preparedness protocol for disasters [18,27,81] How the presence of a caregiver can impact how well older adults do during wildfires [28,76]
COVID-19	2 (2.7)	Wildfires and older adults within the context of COVID-19 [40,47]
Evacuation	26 (34.7)	Individual/community evacuation preparedness and/or experiences (e.g., survival plans, etc.) [27,28,30,40,76,87,96] Medical facilities’ evacuation preparedness and/or experiences [31,33,52,86,97] Needs of/impacts on older adults during evacuations (care disruption, communication, social support, etc.) [18,43,60,79,81,88] Not focused on evacuation, but mention implications, needs, or considerations for evacuation [66,81,84,88]
Health Issues	45 (60)	Wildfire impacts on respiratory and/or cardiovascular health [26,34,54,62,84] Heat-related hospitalizations, illnesses and/or deaths [36,66,80] Effects of wildfires on cancer [81,82] Complications for older adults with dementia [39,43]
Intergenerational	11 (14.7)	How relationships between generations were impacted by wildfires or how these intergenerational relationships could be used as a protective factor against the negative impacts of these natural disasters [30,40] Intergenerational transmission of Indigenous knowledge [55,64,67,68,69,83,94]
Mental Health	19 (25.3)	General discussion of traumatic impact of disasters/wildfires, evacuation, etc. [27,30,40,86,96] Disproportionate impact of disasters/wildfires on older adults’ mental health [18,32,79,81] Vulnerability of individuals with mental health issues during disasters and/or heat [29,32,36,39] Gap in research on mental health impact of disasters/wildfires [81,82]
Morbidity/Mortality	36 (48)	Secondary data of mortality rates related to wildfire smoke-related exposures (PM_2.5,_ PM_10,_ heat, etc.) [26,36,43,51,56,74,85] Secondary data of hospital records measuring morbidity of diseases (respiratory, pulmonary, cardiovascular, cancer, etc.) [37,44,46,48,54,59,60,61,62,81,84,89,90,93,95] Systematic reviews of morbidity and/or mortality from wildfire smoke and related exposures [32,34,57,82,99] Primary data of health impacts related to wildfire smoke and related exposures [45,49,53,78] Future projections of hospital admissions under climate forecasting scenarios [58] General review of climate change impacts on morbidity and mortality [66,72,88]
Social Support or Social Capital	28 (37.3)	Importance of shared social support networks on older adults’ well-being, especially in disasters [19,28,43] Social isolation as a risk factor for older adults during wildfires [18,39,79] Use of social networks on a community level to prepare for and respond to wildfires [47,77,81]

**Table 5 ijerph-20-06252-t005:** Responses or interventions, solutions and recommendations, and future research (*N* = 75).

Categories	*n* (%)	Examples
Problem Description	56 (74.7)	Health impacts of wildfires and related exposures (smoke, heat, etc.) [26,32,41,42,43,44,45,46,51,53,54,56,57,58,59,60,61,62,72,74,78,80,81,82,84,85,88,89,90,93,95,99] Problems/lessons from evacuation and/or disaster response (individuals, communities, and/or facilities) [27,28,40,47,86,87,91,96,97] Disproportionate impact of wildfires and disasters on older adults [18,29,32,39,60,65,66,71,79]
Responses or Interventions	31 (41.3)	Evacuation at individual, organizational, facility, or community levels [27,28,30,31,33,52,86,87] Community-level disaster response (communication, first responders, coordination of services, increasing community-engagement and relationships, etc.) [31,33,47,63,87] Individual level response/interventions (e.g., air filters, masking, survival plans, etc.) [34,41,53,76,84,91,96] Indigenous response/interventions including traditional burning, integrating TEK into conservation/land “management”, employing Indigenous peoples in land “management”, etc. [38,55,64,67,68,69,70,73,75,94,98] Community care, social capital, caring for one another during disasters [27,28,40,50] Interventions by healthcare providers and/or facilities [42,52,86,92,97]
Solutions and Recommendations	61 (81.3)	Need for community-level disaster response workers and coordinators to increase community engagement towards better response and preparedness (community-inclusiveness, responsiveness, education, trust building, outreach, etc. [36,39,43,47,63,77,79,81] Needs for elders leading up to, during, and following evacuation [27,28,43,63,79] Recommendations for healthcare providers and/or facilities regarding clinical or organizational response to wildfires [31,33,66,88,90] Recommendations for public policy [34,60,61,62,71,78,81] Recommendations for utilizing TEK into wildfire management agencies and tactics (in culturally sensitive, ethical ways) [38,67,68,69,70,75,83] Greater need for community care, increased social support, etc. [18,27,28,40,43,47,50,79] Recommendations for climate mitigation and/or drawdown strategies [29,64,71,88,93]
Future Research Directions	61 (81.3)	Future research should work to better understand the impacts of wildfires on the health of older adults [26,31,48,57,65,81] More research is needed on how to develop and evaluate community preparedness and response strategies and the effects of these strategies [27,28,47,84,87,97] Additional research should look at and evaluate effective mitigation strategies [55,63,88] More research should work to find ways to address specific needs of older adults and reduce risks faced by this population before, during, and following wildfires [18,66,77]

## Data Availability

Not applicable. The included literature in references.

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
