# Peer review of "Wildfires and Older Adults: A Scoping Review of Impacts, Risks, and Interventions"

_ijerph, 2023, doi:10.3390/ijerph20136252_

Round 1
Reviewer 1 Report
Summary: This paper is presented as a scoping review of the literature to address a gap in knowledge on disproportionate impacts of wildfire-related disaster events on older adults. The authors searched at least ten databases using clearly laid-out search criteria and terms, and ultimately ended up with 75 papers that they felt warranted more detailed analysis. Contents of the articles were reviewed and then presented in several tables grouping references based on environmental hazard findings, older adult findings, specific topics and themes, and results and interventions. The paper concluded with more in depth discussion of several themes the authors found within the list of references.
Overall I am impressed by the breadth of literature the authors found to discuss in this review. Both the number of databases searched and the specific search terms used seemed more than adequate to make me confident that this reference list covers the current state of the literature. Having the resulting references both listed sequentially at the end of the paper and organized into tables (Tables 2-5) by topics and themes, will be useful for other researchers hoping to make use of the data collected here in their own work.
One major concern I have about this paper is that the overall study goal and research question seem to shift focus between the beginning of the paper and the end. The stated goal at the beginning is to examine a data gap concerning the impacts of wildfire events on older adults, while much of the Discussion section and Conclusions focus heavily on the potential use of older adults as a primary source of knowledge on these impacts, and on specific themes (e.g. TEK, environmental justice) that the authors found to be lacking in the current literature. This change in focus makes the paper overall feel a bit disjointed and confusing to read, as it’s unclear what the authors’ main goal is with this study. This issue could be fairly easily addressed by some rewriting of the Introduction section to more clearly state the intentions of the paper, e.g. addressing gaps in data sources as well as the overall wildfire impact data gap.
Major comments:
· Page 5, Lines 181-190 – It’s unclear to me where this list of topics and themes came from. Were they agreed upon as part of the creation and pilot testing of the data collection tool, or arise later once all the articles had been read?
· Page 14, Lines 363-366 – This reads like a personal opinion of the authors rather than a conclusion of the literature review, even though citations are listed at the end. Maybe add some wording suggesting that the literature suggests this, rather than writing it as a standalone statement.
· Page 16, Lines 374-375 – This is another point at which the purpose of the paper could be clearer
· The overall purpose and goal of this paper is stated several times throughout, but changes somewhat each time.
o Introduction (Page 3, Lines 101-104 ) – “This study seeks to examine [a] gap in literature through a systematic scoping review of scholarly literature to identify and synthesize the existing understanding of the impact of wildfires on older adults” (emphasis mine).
o Discussion (Page 16, Lines 374-375) – “This review systematically synthesized scholarly literature focusing on older adults and wildfires, to help identify priorities and directions for addressing [the data] gap”
o Conclusion (Page 20, Lines 566-568) – “Findings from this scoping review demonstrate how older adults can be an important source of knowledge for wildfire mitigation, response, recovery, and adaptation strategies and should be included in local community planning efforts”
The purposes given in the Discussion and Conclusion sections seem to more closely align with the data as presented in the paper, so I’d suggest a rewrite of the purpose as described in the Introduction to more closely align with these. The study seems to not be searching for consensus in the literature on what the impacts on older adults are, but instead looking at remaining gaps in data and how they might be best addressed.
· Section 4.5 (Elders and Traditional Ecological Knowledge [TEK]) comes a bit out of left field. I realize there’s a lot of discussion within your 75 articles about Indigenous elder knowledge and how it might be used in fire mitigation and management, but that seems well beyond the scope of this paper.
· Page 19, Lines 524-526 – Again, ethics of the use of TEK is a topic well beyond the scope of this paper and appears to be more an (uncited) opinion of the authors versus a finding from the literature review.
Minor comments:
· Page 3, Line 121 – Change “three” to (3) for consistency
· Page 3, Lines 106 and 108 – These sentences are structured very similarly, consider rewriting one of them.
· Page 4, Line 169 – Move the sentence beginning “Data collected on each article…” to the beginning of this section to make it clear what the “data collection tool” is being used for.
· Page 7, Table 1 – the last item (Not Applicable) has a description that cuts off at the end
· Page 8-10, Table 2 – Formatting issues in the first column; many headings are cut off
· Page 11, Line 300 – Consider saying “may not be” rather than “are not”
· Lines 175-189 – This list of topics/themes should be listed in the same order as the tables presented. For example, “questions related to how [needs to be included] older adults are included is listed as the second topic, but it is the third table show.
· Line 324 – Can you be more descriptive here? What do you mean “focused a problem description”? It might be helpful to describe how you divided up this category (e.g. studies of impacts, studies on responses/interventions, etc).
· Line 327 – PM2.5 needs to be defined
Author Response
Please find detailed responses, attached.

Reviewer 2 Report
This review examined literature to explore the impact of wildfires on older adults. In general, the authors have collected abundant literature and extract key words by different topics. Thus, this review has replied to the global concern of impact of wildfire on vulnerable population timely with valuable public health implications. Yet there are some suggestions need to be addressed.
1. Some (may be many) studies cited from the review have examined the degree of impact of wildfire on older adults, e.g., using Odds Ratio or Relative Risk. If the results from different literature in any topic (e.g., Older Adult Findings) are consistent. The authors need to list some of the important results (data) from original research articles if it is possible.
2. The strengths and limitations of the review need to be clearly addressed.
The manuscript needs minor language corrections. E.g., im some places, PM2.5 should be revised to PM2.5. Line 5: "Graduate School of Social of Social Work" should be "Graduate School of Social Work".
Author Response

(The authors gave the same response as above.)
